# Boosting Inference Efficiency: Unleashing the Power of Parameter-Shared Pre-trained Language Models

**Weize Chen**[1][*], **Xiaoyue Xu**[1][*], **Xu Han**[1][†], **Yankai Lin**[2],
**Ruobing Xie**[2], **Zhiyuan Liu**[1][†], **Maosong Sun**[1], **Jie Zhou**[3]

[1]NLP Group, DCST, IAI, BNRIST, Tsinghua University, Beijing
[2]Gaoling School of Artificial Intelligence, Renmin University of China, Beijing
[3]Pattern Recognition Center, WeChat AI, Tencent Inc.
chenwz21@mails.tsinghua.edu.cn
hanxu2022@tsinghua.edu.cn

## Abstract

Parameter-shared pre-trained language models (PLMs) have emerged as a successful approach in resource-constrained environments, enabling substantial reductions in model storage and memory costs without significant performance compromise. However, it is important to note that parameter sharing does not alleviate computational burdens associated with inference, thus impeding its practicality in situations characterized by limited stringent latency requirements or computational resources. Building upon neural ordinary differential equations (ODEs), we introduce a straightforward technique to enhance the inference efficiency of parameter-shared PLMs. Additionally, we propose a simple pre-training technique that leads to *fully* or *partially* shared models capable of achieving even greater inference acceleration. The experimental results demonstrate the effectiveness of our methods on both autoregressive and autoencoding PLMs, providing novel insights into more efficient utilization of parameter-shared models in resource-constrained settings.

## 1 Introduction

In recent years, there has been a significant increase in the number of parameters of PLMs. This began with the advent of BERT (Devlin et al., 2019), containing 340 million parameters, and has escalated to models like T5 (Raffel et al., 2020), GPT-3 (Brown et al., 2020), and PALM (Chowdhery et al., 2022), with the latter reaching an astounding 540 billion parameters. The trend of PLM expansion has, undeniably, improved performance across numerous tasks. Nonetheless, the corresponding increase in computation and storage requirements has raised substantial barriers for scenarios characterized by stringent latency requirements or resource limitations. While PLMs encompassing merely a

few billion parameters such as LLaMA (Touvron et al., 2023), Vicuna (Chiang et al., 2023), and Alpaca (Taori et al., 2023) have exhibited remarkable capabilities, their application remains constricted in numerous resource-constrained environments.

In contrast to the monumental advances in PLMs, the real-world applications often still favor more established models such as BERT and GPT-2 (Radford et al., 2019). These models, despite their relatively fewer parameters, deliver satisfactory performance across many tasks while requiring significantly less resources. This balance offers an appealing trade-off between performance and cost. Moreover, *parameter sharing* techniques have successfully demonstrated that model size can be greatly reduced without significant performance degradation, mitigating the storage burden and yielding better cost-effectiveness. This has sparked an interest in parameter-shared PLMs (PSPLMs) like AL-BERT (Lan et al., 2020), a derivative of the BERT architecture that shares parameters across all layers, effectively reducing model size and memory requirements. Still, it's critical to recognize that parameter sharing alone doesn't guarantee reduced inference time since the number of layers processed during each forward pass remains unchanged. In other words, while it resolves the storage issue, it does not address the computational challenge.

Early exit techniques promise to reduce the number of layers processed during inference by halting computation at early layers (Zhou et al., 2020; Wang et al., 2022; Schuster et al., 2022). While effective, these methods typically require additional trained classifiers or computationally expensive dot products between the vocabulary matrix and the hidden states at each layer. This circumstance prompts the question: Can a method be proposed to reduce the inference cost *without introducing extra modules or computations*, and could it be *complementary* to early exit techniques, allowing their combined use for further acceleration?

---

[*]Equal Contribution
[†]Corresponding author.

In this study, we show that the problem can be well addressed in PSPLMs. Specifically, we illustrate how significant acceleration in PSPLM inference can be achieved through our straightforward yet effective technique. This technique, inspired by the principles of neural ODEs, accelerate the inference without necessitating the addition of modules or calculations to the model. Hence, in addition to the inherent storage efficiency, our method notably makes PSPLMs to be computational efficient. We also introduce a pre-training method for PSPLMs with a slightly altered forward propagation rule. Experiments reveal that our proposed pre-training method prepares the model for even greater acceleration during inference, and we give theoretical explanation to aptly support our method.

We further extend the application of our method beyond the domain of fully shared PSPLMs. Our research demonstrates the potential of our acceleration strategy in the context of more complex and capable partially-shared PLMs, and even hints at its applicability to unshared models. This broader applicability shows the flexibility and robustness of our approach. Additionally, our method is not in competition with other acceleration strategies. We demonstrate that our method can be combined orthogonally with early exit techniques, thereby facilitating further acceleration. Remarkably, this synergy of methods makes the partially-shared model surpass its unshared equivalent within an equivalent computational budget.

In essence, our work offers a novel route to accelerate the inference for PSPLMs, and lay a novel foundation for unleashing the PSPLMs' potential, offering critical insights for deployment of PSPLMs in resource-constrained settings.

## 2 ODE Perspective on Residual Networks

We begin by providing a brief overview of the relationship between residual networks and ODEs, which forms the fundamental basis of our research.

In a $T$-layer residual network, we denote the layer $t$ as $f_{\theta_t}$. The update formulation for the hidden state $h$ can be expressed as:

$$h_{t+1} = h_t + f_{\theta_t}(h_t). \qquad (1)$$

Remarkably, this update scheme aligns with Euler's method for solving ODEs. Consider an ODE:

$$y(0) = y_0, \quad y'(t) = f_t(y(t)), \qquad (2)$$

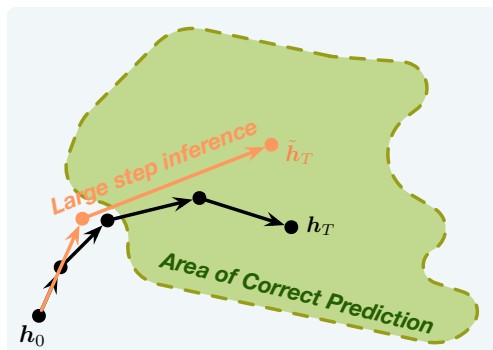

Figure 1: An illustration of reducing the iteration count during inference by enlarging the step size.

Euler's method approximates the solution of this ODE from $t = 0$ to $T$ by dividing the interval into $n$ steps, each with step size $s_i$, such that $\sum_{i=0}^{n-1} s_i = T$. The method iteratively computes the following formula from $i = 0$ to $n - 1$:

$$y_{i+1} = y_i + s_i \cdot f_t(y_i). \qquad (3)$$

The final value $y_n$ serves as an approximation of the solution to Eq. 2 at time $T$. The correspondence between Eq. 1 and Eq. 3 is evident. A $T$-layer residual network can be interpreted as parameterizing the vector field $f$ that characterizes the derivative along the path from the input space to the final output space. The ODE perspective generalizes the concept of depth in residual networks to the continuous domain, where the notion of progression from input to output is captured by the continuous *time* rather than discrete depth or layer index.

During the inference process of a trained model, the vector field remains fixed because the parameters are frozen. As a result, the model's inference can be seen as solving an ODE within this vector field using Euler's method, where the initial value corresponds to the input embedding, and the solution time is $T$. Furthermore, the pre-norm Transformer architecture is also a type of residual network (see Appendix A). Therefore, the ODE perspective we have presented can be applied to the pre-norm Transformer architecture as well.

## 3 Method

### 3.1 Enlarging the Step Size

In the process of solving the ODE, the choice of the step size $s_i$ in Eq. 3 has a significant impact on the speed and accuracy of the solution. Given the final time $T$, a larger step size reduces the number of iterations, resulting in faster computation but decreased accuracy. This trade-off allows us to

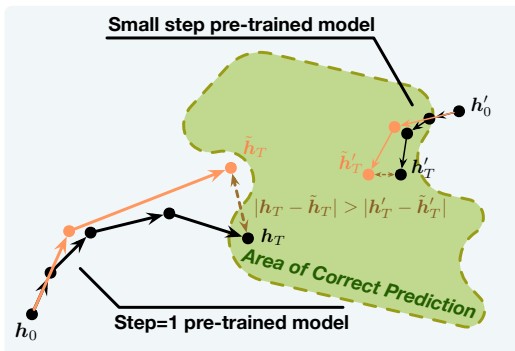

Figure 2: An illustration of the difference between the models pre-trained with different step size.

sacrifice a certain degree of solution accuracy in exchange for inference speed. To expedite model inference, we propose employing a larger step size during inference compared to training (Fig. 1).

Specifically, for PSPLMs, the vector field $f$ at any given time $t$ is parameterized by exactly the same set of parameters. Consequently, the time dependence in $\theta_t$ (as shown in Eq. 1) can be omitted, leading to the forward rule $\boldsymbol{h}_{t+1} = \boldsymbol{h}_t + f_\theta(\boldsymbol{h}_t)$, where $\theta$ now represents the shared layer parameters. By applying different scaling factors $\beta_t > 1$ to the original step size $s = 1$ at different layers, and reducing the number of layers such that $\sum_t \beta_t \approx T$, the model still mathematically solves the ODE from $t = 0$ to $T$, albeit using larger step sizes $\beta_t s = \beta_t$ at each layer. The updated forward rule can be written as:

$$\tilde{\boldsymbol{h}}_{t+\beta_t} = \tilde{\boldsymbol{h}}_t + \beta_t \cdot f_\theta(\tilde{\boldsymbol{h}}_t), \quad \beta_t > 1. \quad (4)$$

In practice, we perform minimal search to determine a set of suitable $\{\beta_t\}$ values. In Section 4.2, we will demonstrate that by simply changing the forward rule to Eq. 4, the inference of existing PSPLMs can already be accelerated while maintaining overall performance to a satisfactory extent.

## 3.2 Pre-Training with Smaller Step Size

Ideally, if the approximate result $\tilde{\boldsymbol{h}}_T$ obtained using scaled-up step sizes is close to the result $\boldsymbol{h}_T$ obtained with the original step size, the overall performance can be greatly preserved. Conventionally, pre-training of PSPLMs is conducted with a step size of $s = 1$. However, from a theoretical standpoint, the error analysis of Euler's method suggests that selecting a smaller step size $s$ during pre-training may enable better acceleration during inference. Under certain mild conditions, we prove in Appendix B the following inequality:

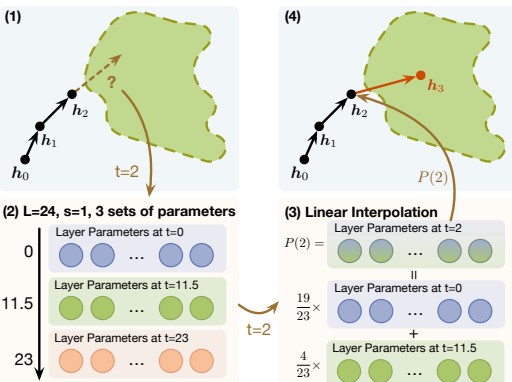

Figure 3: An illustration of our partially-shared model. The example shows how a model with number of layers $L = 24$, step size $s = 1$ and $n = 3$ sets of parameters determines the layer parameters for $t = 2$.

$$\|\tilde{\boldsymbol{h}}_T - \boldsymbol{h}_T\| \leq K(1 + \beta^*)s, \quad (5)$$

where $\beta^*$ is the largest scaling factor used across all the layers, and $K$ is a constant determined by the model parameters. This inequality indicates that the difference between $\tilde{\boldsymbol{h}}_T$ and $\boldsymbol{h}_T$ is bounded to the magnitude of the largest scaled-up step size $\beta^* s$ employed during inference. Assuming that the value of $K$ is approximately the same for models pre-trained with different step scales, then when the step sizes are scaled up by the same factors, the model pre-trained with a smaller $s$ produces a final approximated hidden state that is closer to the hidden state obtained using its original step size.

Empirically, we will show in Section 4.3 that PSPLMs pre-trained with a reasonably small step size can achieve improved performance when reducing the number of iterations during inference.

## 3.3 Generalizing to Partially-Shared PLMs

Shared-parameter models employ the same set of parameters to parameterize the derivative at *different* discrete time steps during training. This property offers the models the ability to generalize from discrete time to continuous time during inference (Eq. 4). On the other hand, unshared models use distinct parameters to parameterize the derivative at each discrete time step, making it challenging to apply a continuous scaling factor $\beta$ to the step size during inference. For instance, if we use $\beta_0 = 1.3$ and $s = 1$, the model would need to provide the derivative at $t = 1.3$ at the next iteration, which is unattainable as the unshared model can only provide the derivative at $t = 1$ using layer 2 or at $t = 2$ using layer 3, but not at any intermediate time.

However, we will demonstrate that pre-training a partially-shared PLM with time-dependent parame-

ters represented by a piece-wise linear function can enhance the model's capabilities while benefiting from the accleration method we introduce in Section 3.1. Given a language model with $L$ layers and step size $s$, and $n$ ($2 \leq n \leq L$) sets of layer parameters, denoted as $\theta = \{\theta_0, \theta_1, ..., \theta_{n-1}\}$, we uniformly position the $n$ sets of parameters within the range from $0$ to $(L-1)s$, with each interval spanning $\Delta = (L-1)s/(n-1)$.

To determine the parameters at a specific time, denoted as $t$, we define the function $P(t)$ that returns the parameters at time $t$. We first identify the left and right boundary indices of the interval in which $t$ resides, denoted as $l$ and $r$, respectively. Subsequently, we perform linear interpolation between $\theta_l$ and $\theta_r$ to obtain the parameters at time $t$. $P(t)$ can be formally written as:

$$\Delta = \frac{(L-1)s}{n-1}, \quad l = \lfloor \frac{t}{\Delta} \rfloor, \quad r = \lceil \frac{t}{\Delta} \rceil,$$
$$P(t) = \theta_l + \frac{t - l\Delta}{\Delta}(\theta_r - \theta_l). \qquad (6)$$

Fig. 3 shows an example of this process. The model is referred to as *partially-shared* since it does not share all layer parameters; instead, it shares a set of parameters and uses interpolation to obtain the parameters at different time. Notably, when $n = L$, it becomes the unshared model, and the fully-shared model is a special case if we allow $n = 1$.

By learning to use linear interpolation to derive parameters for different time steps during training, the model can naturally generalize to the continuous domain and provide derivatives for any time step during inference. We will demonstrate at Section 4.4 that these partially-shared PLMs exhibit notable advantages, including better or comparable performance to their unshared counterparts, as well as the ability to enable accelerated inference through a scaled-up step size.

## 4 Experiments and Analyses

### 4.1 Experimental Setups

We investigate the effectiveness of our suggested inference acceleration technique on both autoregressive and autoencoding models. Specifically, we pre-train GPT-2$_{\text{large}}$ models and pre-norm BERT$_{\text{large}}$ models both with shared parameters under diverse settings, as elucidated in the subsequent sections. All GPT-2 models are pre-trained on the OpenWeb-Text dataset (Radford et al., 2019), and all BERT models are pre-trained on the Pile dataset (Gao

et al., 2021). Detailed information on hyper-parameters and pre-training configurations can be found in Appendix D. It is important to mention that while we exclusively focus on parameter sharing among layers, our proposed method can be seamlessly incorporated alongside other parameter-reduction techniques such as embedding factorization used in ALBERT.

For downstream task evaluation, we measure the zero-shot perplexity (PPL) on Wikitext-103 (Merity et al., 2017), and zero-shot accuracy on LAMBADA (Paperno et al., 2016) for GPT-2 models. And as for BERT models, they are fine-tuned on different tasks including MNLI, SST-2 (Wang et al., 2019), RACE (Lai et al., 2017), SQuAD and SQuAD2.0 (Rajpurkar et al., 2016) separately. Configuration details and metrics for these downstream tasks can be found in Appendix E. During the inference, we experiment with different iteration counts and for each count, we perform a minimal search on the $\beta$ for each layer within the set $\{1.0, 1.1, \ldots, 3.0\}$ using Optuna (Akiba et al., 2019), and report the best results. Unless explicitly stated otherwise, both BERT and GPT-2 models mentioned hereafter are parameter-shared.

### 4.2 Inherent Highways: Scaling Up Step Sizes

As described in Section 3.1, from the perspective of ODEs, we can naturally accelerate PSPLMs by increasing the step sizes. In other words, there may be *inherent highways* in PSPLMs, and we may utilize them by increasing the step size and decreasing the number of iterations.

To validate the presence of these inherent highways in PSPLMs, we pre-train GPT-2$_{\text{large}}$ and BERT$_{\text{large}}$ models under the conventional setting (i.e., $s = 1$), and evaluate their inference performance on a variety of downstream tasks with different iteration counts and step sizes. The results are shown in Table 1. Additionally, we compute relative changes in performances for reduced iterations as $\frac{p_{\text{reduced}} - p_{\text{orig}}}{p_{\text{orig}}}$, where $p$ represents the performance, and we report these values in parentheses.

Our experimental results reveal that a clever reduction in the iteration count presents an opportunity for substantial computational savings while maintaining most of the model performance. When the iteration count decreases from 24 to 20, the performance impact across all datasets is virtually negligible. For BERT, variations in performance are consistently contained within a margin of $\pm$

| #Iters | BERT | | | | | | GPT-2 | | |
|---|---|---|---|---|---|---|---|---|---|
| | Speed | MNLI↑ | SST-2↑ | RACE↑ | SQuAD↑ | SQuAD2↑ | Speed | Wiki-103↓ | LAMBADA↑ |
| 24 | 1.00x | 83.6 | 91.1 | 64.3 | 90.2 | 81.5 | 1.00x | 33.0 | 31.1 |
| 20 | 1.20x | 83.6 (+0.0%) | 90.8 (-0.3%) | 64.1 (-0.3%) | 90.0 (-0.2%) | 81.5 (+0.0%) | 1.16x | 33.5 (+1.4%) | 29.6 (-5.0%) |
| 16 | 1.47x | 83.3 (-0.4%) | 90.9 (-0.1%) | 62.9 (-2.2%) | 89.3 (-1.0%) | 80.3 (-1.5%) | 1.40x | 35.3 (+7.0%) | 30.9 (-0.8%) |
| 12 | 1.92x | 81.1 (-3.0%) | 90.4 (-0.8%) | 59.4 (-7.7%) | 83.0 (-8.0%) | 65.0 (-20.2%) | 1.77x | 105.1 (+218.4%) | 5.3 (-83.1%) |

Table 1: Inference performance of PSPLMs pre-trained with step size 1. Values in parentheses indicates relative change from non-reduced iteration counts. Speed reflects the acceleration of the forward pass wallclock time.

0.3% across all tasks. Simultaneously, the GPT-2 model shows only a slight increase in perplexity on Wikitext-103 from 33.0 to 33.5. Even with a further reduction in iteration count to 16, the models continue to deliver respectable performance. For BERT, the majority of tasks report a minimal performance decrease, with the highest decrease appearing in the RACE task at -2.2%. For GPT-2 model, although the perplexity on Wikitext-103 increases to 35.3 and the accuracy on LAMBADA decreases to 30.9, the performance still stays within an acceptable range.

Overall, these results suggest that the step size per iteration can be scaled up and leads to a reduction in the number of iterations in conventional PSPLMs without a significant compromise on performance. In essence, our approach enables a computational reduction by approximately $1/3$ to $1/6$. However, further reductions in iteration does result in performance degration, which can be addressed in the subsequent section.

### 4.3 Acceleration Boost: Mini-Step Pretraining

In Section 3.2, we posited that pre-training PSPLMs with small step sizes may make the models more conducive to acceleration during the inference. This section provides empirical validation of these theoretical insights.

#### 4.3.1 Performance Across Downstream Tasks

For a fair comparison, we maintain identical pre-training configurations and data and train 4 models with step size 1, 0.1, 0.05, and 0.01 respectively. The performance is presented in Fig. 4, where we have several noteworthy observations:

**Small step sizes do not detrimentally affect performance within a reasonable range.** We first look at the performances when the iteration count is not reduced (24 in the figures). BERT models pretrained with smaller step sizes demonstrate comparable, and in some instances superior, performance on various downstream tasks in comparison to the conventionally pretrained BERT. The only exception is the MNLI task, where the latter model

performs marginally better. However, it should still be noted that extremely small step size of 0.01 still negatively impacts the model's performance across all tasks. But overall, a reasonably small step size does not impact the model's capacity.

**Small step sizes enhance performance retention when reducing the iteration count.** Looking at the performance at 12 iterations across models with varying step sizes, as expected, we generally observed a decline in comparison to the performance attained at 24 iterations. However, models with smaller step sizes exhibit remarkably better performance retention. Particularly noteworthy is GPT models pretrained with small step sizes, as they exhibit a significantly better zero-shot performance retention on both the LAMBADA and Wikitext-103 tasks. Notably, as we do not inroduce any additional computational overhead, the speedups of BERT and GPT are the same as those reported in Table 1, which is almost linear to the reduced number of iteration, while the performance retention is significantly improved.

**Reducing iteration count enhances performance on certain datasets.** This finding aligns with observations made in earlier studies on early exits, suggesting that preventing models from *overthinking* can enhance both accuracy and robustness (Zhou et al., 2020; Balagansky and Gavrilov, 2022). However, unlike these previous studies, our approach demonstrates that we can effectively and easily prevent overthinking for models pretrained with smaller step sizes without auxiliary modules.

#### 4.3.2 Analyzing the Possible Mechanism

We further explore why models pretrained with a small step size result in PSPLMs that are more efficiently accelerated during inference. Our analysis reveals two main advantages for models pretrained with smaller step sizes:

**Reduced absolute and relative difference when the iteration count is decreased.** We decrease the iteration count for all models to 20, 16, and 12, use the searched step scales and calculate the absolute difference, denoted as $\|\boldsymbol{h}_T - \tilde{\boldsymbol{h}}_T\|$,

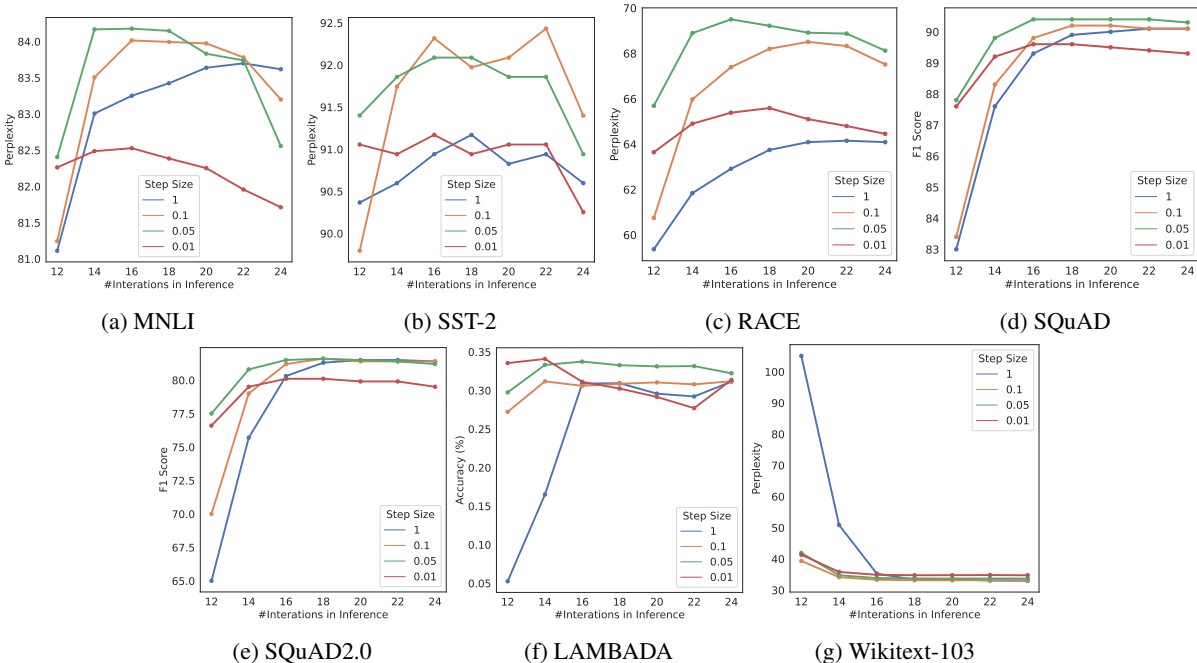

|     |     |     |     |
| --- | --- | --- | --- |
| (a) MNLI | (b) SST-2 | (c) RACE | (d) SQuAD |

|     |     |     |
| --- | --- | --- |
| (e) SQuAD2.0 | (f) LAMBADA | (g) Wikitext-103 |

Figure 4: The inference performance of parameter-shared models pre-trained with different step size. (a-e) The accuracy of BERT on MNLI, SST-2 and RACE, and F1 score on SQuAD and SQuAD2.0. (f-g) The zero-shot accuracy of GPT-2 on LAMBADA, and the zero-shot perplexity of GPT-2 on Wikitext-103.

and the relative difference, expressed as $\|\boldsymbol{h}_T - \tilde{\boldsymbol{h}}_T\|/\|\boldsymbol{h}_T\|$. Here we keep the notations consistant with Eq. 5. These values represent the difference between the approximated and the original final hidden states.

As demonstrated in Fig. 10, the final hidden state approximation from a model pre-trained with a smaller step size presents a closer resemblance to the original final hidden state, both in terms of absolute and relative difference. These observations suggest that when the number of iterations is decreased, models pre-trained with smaller step sizes could yield results that align more closely with those from models with unreduced iterations on most tasks, thereby better preserving performance.

**Enhanced smoothness in the vector field.** To further our analysis, we compute the cosine similarity between the derivatives $f(x)$ produced by the model at two consecutive iterations, denoted as $\text{CosSim}\left(f(\boldsymbol{h}_i), f(\boldsymbol{h}_{i-1})\right), i \in 1, 2, \ldots, 23$. The results are plotted in Fig. 5.

Figs. 5 and 8 reveal an increasing trend of cosine similarity as the layer index increases, with smaller step size generally resulting in higher cosine similarity in the early layers. Although a step size of 0.1 also appears to have lower similarities for the first few layers, there is a swift increase as the layer index increases. The cosine similarities for step sizes of 0.01 and 0.05 consistently remain

over 0.8 across all layers, suggesting an almost parallel alignment of the derivatives at different time, that is, a smoother vector field. In other words, the paths from the input embedding to the final output are more "straight" for models pre-trained with small step size, thus allowing us to reduce the number of iteration and enlarge the step size during inference.

### 4.4 Expanding Horizons: Partial Sharing

In this section, we pre-train partially-shared PLMs, and apply our method described in Section 3.3 to them. This experimental validation substantiates the possibility of inference acceleration in more complex, partially-shared, and even unshared models.

For BERT_large and GPT-2_large, we conduct pre-training with $n = 12$ sets of parameters and step sizes of 0.1 and 0.05, respectively. We establish baselines by pre-training unshared BERT and GPT models using the equivalent configurations, except that the total number of layers is set to 12. This ensured the same number of parameters between our partially-shared models and the baseline models.

The results of downstream tasks are illustrated in Figs. 6 and 9. The unshared model performance is represented by the red dashed line in each figure. As anticipated, partially-shared models, benefiting from the increased number of parameters, signifi-

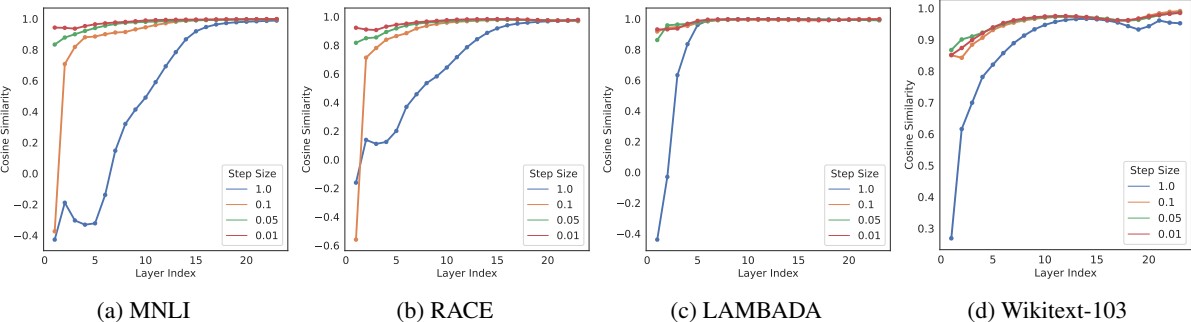

(a) MNLI     (b) RACE     (c) LAMBADA     (d) Wikitext-103

Figure 5: The cosine similarity between the derivatives given by the model at two consecutive iterations.

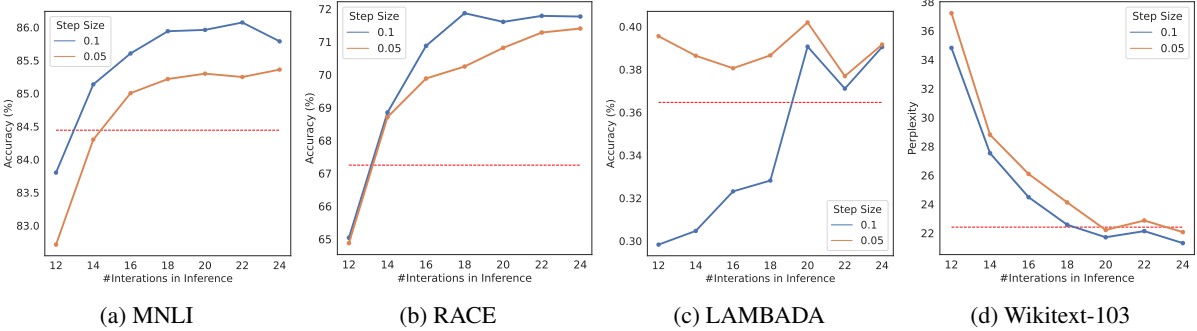

(a) MNLI     (b) RACE     (c) LAMBADA     (d) Wikitext-103

Figure 6: The inference performance of partially parameter-shared models with $n = 12$ sets parameters. The red dashed line represents the performance of the unshared 12 layer model.

cantly outperform those fully-shared counterparts in Section 4.3. Furthermore, because they have a larger iteration count, they outperform unshared models with the same parameter count. The results show the feasibility of pre-training partially-shared PLMs via linear parameter interpolation.

Our focus, however, is on the performance *post* reduction in iteration count. At 14 iterations, BERT pre-trained with a step size of 0.1 either surpasses or matches the unshared 12-layer baseline across all tasks. However, when the iteration count is further decreased to 12, a performance drop is observed, making it marginally underperform the baseline. Nonetheless, it still achieves over 98% of the baseline performance in most tasks. As for GPT, the model pre-trained with a step size of 0.05 exhibits impressive performance retention on LAMBADA across all iteration counts, consistently beating the unshared baseline. Although perplexity on Wikitext-103 rises with reduced iterations, it remains at an acceptable level.

It is crucial to underline that this iteration reduction is achieved *without* additional training post pre-training and fine-tuning. While the partially-shared model may lag behind the unshared baseline in some tasks post-reduction, it stays competitive, which is a non-trivial achievement considering that we merely increased the step size during inference.

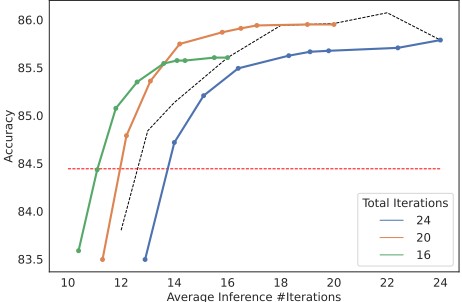

Figure 7: Performance of early-exit BERT on MNLI. The black dashed line represents partially-shared BERT pre-trained with $s = 0.1$, while the red dashed line denotes the unshared 12-layer BERT.

We have also tried the $n = 24$ setting, which is equivalent to the unshared 24-layer model, the performance retention is satisfactory when the scaling factors are integers. We place the results and analysis in Appendix G.

### 4.5 Rapid Inference: Early Exit Integration

This section aims to demonstrate the potential for combining our approach with the early exit technique to further enhance model performance under reduced iteration counts. We adopt the strategy employed by DeeBERT (Xin et al., 2020), in which internal classifiers are trained at every layer of a frozen, fine-tuned BERT model to predict the final label. Similarily, we train classifiers on the partially-shared BERT that has been pre-trained

with $s = 0.1$. See Appendix F for more details.

Initially, we conduct a step scale search for the model at 24, 20, and 16 iterations, as we have done in Section 4.4. Subsequently, classifiers are trained on these models with their respective iteration counts and searched step scales. During inference, the entropy of the prediction distribution at each layer guide the decision to halt. By adjusting the entropy thresholds, we can manage the trade-off between performance and efficiency.

The results are presented in Fig. 7. Evidently, when the DeeBERT technique is applied to models with reduced iteration counts of 16 and 20, it succeeds in outperforming the unshared 12-layer model under the equivalent computational budget. This finding indicates that the early exit strategy and reduction of iteration count using large step scales are indeed synergistic. Their integration effectively bolsters performance retention, yielding a remarkable inference acceleration.

## 5 Related Work

**PSPLMs.** The increasing size and memory usage of PLMs have prompted research efforts focused on parameter sharing in these models. Several approaches have been proposed, demonstrating the potential to maintain comparable performance while significantly reducing the model size. Universal Transformers (Dehghani et al., 2019) and ALBERT (Lan et al., 2020) share parameters across all layers. Takase and Kiyono (2021) propose to share parameters for every two consecutive layers or share layer parameters cyclically, while Xue et al. (2022) propose to share all layer parameters except bias terms and layer normalization modules. These advanced strategies enhance model capacity at the cost of increased parameter number. However, none of these methods reduce the computational cost during inference: the computations required for the inference of shared and unshared PLMs are still identical.

**Early Exit.** The efficiency of inference in PLMs has become a significant concern for deployment, leading to extensive research efforts focused on inference acceleration. Early exit techniques aim to terminate the inference process in early layers and bear close relevance to our work. Many early exit methods necessitate an internal classifier to be applied to the intermediate hidden states of the early layers, thus requiring joint training with the PLMs themselves (Zhou et al., 2020; Wang et al., 2022),

or training as a separate stage with the PLMs held frozen (Xin et al., 2020; Liu et al., 2020). Our method distinguishes itself by reducing the number of layers during inference without the need for an additional classifier that requires training. Also, our method is complementary to early exit technique, and can be jointly leveraged to accelerate the inference.

**Neural ODEs.** The connection between residual networks and ordinary differential equations has been extensively explored in prior research (E, 2017), where different designs of residual networks can be linked to diverse numerical discretizations of ODEs (Chen et al., 2018; Lu et al., 2018). Neural ODEs extend the concept of residual networks to continuous-depth architectures. In our work, we build upon the ODE perspective of residual networks and propose to accelerate the PSPLMs by increasing step size, and from the error analysis of Euler's method, we propose a simple pre-training technique to enable further inference acceleration.

**Hyper-Networks.** We adopt the linear interpolation of a piece-wise linear function as parameters for different layers to build partially-shared PLMs. This bears resemblance to hypernetworks (Ha et al., 2017), where the parameters of a neural network are generated by a specific function or another neural network. The parameterization of model parameters in a hypernetwork style has found wide application in various domains, including neural architecture search (Brock et al., 2018), meta-learning (Requeima et al., 2019), and neural ODEs (Chen et al., 2018).

## 6 Conclusion

In this study, we draw inspiration from the ODE perspective on residual networks. Our research proposes straightforward strategies to expedite the inference process for both fully and partially-shared PLMs. The results of our work reveal that PSPLMs can not only reduce the storage and memory costs, but also reduce the time costs. Furthermore, when our approach is coupled with the early exit technique, the partially-shared PLMs demonstrate superior performance compared to unshared models under the same computational budget. We believe that our methodology harbors substantial potential, particularly in the acceleration of inference in unshared PLMs - a promising avenue for future research. We anticipate the extension of our tech-

niques to the acceleration of large language models encompassing billions of parameters, and look forward to further explorations in this field.

## Acknowledgements

This work is supported by the National Key R&D Program of China (No.2022ZD0116312), National Natural Science Foundation of China (No. 62236004) and Institute Guo Qiang at Tsinghua University.

## Author Contribution

In the preparation and discussion of the project, Weize Chen, Xiaoyue Xu, Yankai Lin designed the algorithm. Weize Chen and Xiaoyue Xu wrote the code and conducted the experiments. Weize Chen and Xiaoyue Xu wrote the initial draft. Xu Han, Yankai Lin, Ruobing Xie, and Zhiyuan Liu significantly edited and improved the paper. Maosong Sun and Jie Zhou provided valuable advice to the research.

## Limitations

The effect of a larger step size on model's inference performance could vary across different models and tasks. Although we have tried to include different types of downstream tasks and different types of models to show the generalizability of our method, it could still fail on some certain situations. Morevoer, while we show that our method can be applied to partially-shared PLMs, its effectiveness in accelerating unshared PLMs remains to be further explored. We have only conducted some basic experiments on the unshared model to show the potential of the method. Further research is needed to determine if and how our method can be adapted for unshared models.

## Ethics Statement

This work focuses on the acceleration of inference in PSPLMs. While our research does not directly involve human subjects or sensitive data, it does have implications for the broader use of these models in society. The primary potential ethical impact of our work involves the expanded use of PLMs. By providing methods for accelerating PSPLMs, we may enable wider deployment of these models, including in contexts with limited computational resources. While this has potential benefits, such as increased accessibility to advanced language processing technology, it may also have unintended consequences. For example, accelerated PLMs may be used to produce fake text more efficiently, potentially contributing to misinformation or fraud.

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

# A  Proof on Pre-Norm Transformer Architecture as a Residual Network

To demonstrate that the pre-norm Transformer architecture can be considered a residual network, we will analyze the computation within each layer. Let us consider the hidden state $\boldsymbol{h}_t$, and examine the computation in the $t$-th layer of the pre-norm transformer, which can be represented as follows:

$$\boldsymbol{x}_t = \text{ATT}_t\big(\text{LN}_t^1(\boldsymbol{h}_t)\big) + \boldsymbol{h}_t, \qquad (7)$$

$$\boldsymbol{h}_{t+1} = \text{FFN}_t\big(\text{LN}_t^2(\boldsymbol{x}_t)\big) + \boldsymbol{x}_t. \qquad (8)$$

By substituting Eq. 7 into Eq. 8, we obtain:

$$\begin{aligned}\boldsymbol{h}_{t+1} &= \boldsymbol{h}_t + \text{FFN}_t\big(\text{LN}_t^2(\boldsymbol{x}_t)\big) + \text{ATT}_t\big(\text{LN}_t^1(\boldsymbol{h}_t)\big) \\ &= \boldsymbol{h}_t + f_t(\boldsymbol{h}_t).\end{aligned}$$

Hence, we observe that the pre-norm Transformer architecture functions as a residual network, where each layer parameterizes the derivatives of the hidden states and updates the hidden states utilizing Euler's method.

# B  Proof on the Errors of Euler's Method

Here we provide the proof in the case where the dimension of state is 1 for simplicity. It can be easily generalized to high-dimensional setting. At time $t_0$, we denote the real state of the ODE as $h(t_0)$. The Euler's method (or residual network) approximates the real state at $t_1 = t_0 + s$ as

$$h_{t_1} = h(t_0) + s \cdot f\left(h(t_0), t_0\right), \qquad (9)$$

here we move the notation of time $t$ from the subscript (Eq. 3) into the parentheses to explicitly denote the dependency on time $t$.

The real state $h(t_1)$ can be expanded by using the Taylor series expansion around $t_0$:

$$h(t_1) = h(t_0) + sh'(t_0) + \frac{s^2}{2}h''(\tilde{t}_0), \quad (10)$$

where $\tilde{t}_0$ is some number between $t_0$ and $t_0 + s$, and $h'(t_0) = f(h(t_0), t_0)$.

**Local Truncation Error.**   Now we derive the local truncation error $T$ of Euler's method, which is essentially the discrepancy between the real state $h(t_1)$ and the one-step approximated state $h_{t_1}$ start from $h(t_0)$. It is formally written as:

$$T_{t_1} = h(t_1) - (h(t_0) + s \cdot f(h(t_0), t_0))$$
$$= \frac{s^2}{2}h''(\tilde{t}_0). \quad (11)$$

We can further derive the $h''(t_0)$ by differentiating $h'(t_0) = f(h(t_0), t_0)$:

$$h''(t_0) = f_h(h(t_0), t_0)h'(t_0) + f_t(h(t_0), t_0)$$
$$= f_h(h(t_0), t_0)f(h(t_0), t_0) + f_t(h(t_0), t_0). \quad (12)$$

Assume that $f$ and its derivatives $f_h(h(t_0), t_0)$ and $f_t(h(t_0), t_0)$ are all continuous and bounded, then there exists a constant $M$ so that

$$|h''(t_0)| \leq M. \quad (13)$$

Taking Eq. 13 into Eq. 11, then we can establish the local truncation error of Euler's method:

$$T_{t_1} \leq \frac{M}{2}s^2. \quad (14)$$

Since $M$ is a constant, the local truncation error of Euler's method is of order $O(s^2)$, i.e., the square of the step size.

**Global Truncation Error.**   The global truncation error is the accumulated error from initial time $t_0$ to final time $T$. To derive the global truncation error, we first define $e_{t_i} = h(t_i) - h_{t_i}$ as the difference between the real state $h(t_i)$ and approximated state $h_{t_i}$ at time $t_i$. Recall that we have

$$h(t_{i+1}) = h(t_i) + s \cdot f(h(t_i), t_i) + T_{t_i}, \quad (15)$$
$$h_{t_{i+1}} = h_{t_i} + s \cdot f(h_{t_i}, t_i), \quad (16)$$

where Eq. 16 is the Euler's method, and Eq. 15 is the Euler's method with local truncation error term. Substracting Eq. 16 from Eq. 15, we have

$$e_{t_{i+1}} = e_{t_i} + s(f(h(t_i), t_i) - f(h_{t_i}, t_i)) + T_i. \quad (17)$$

From Eq. 14, we have $|T_{t_i}| \leq \frac{M}{2}s^2$. By substituting it into Eq. 17 and applying the absolute value inequality, we can see that

$$|e_{t_{i+1}}| \leq |e_{t_i}| + s|f(h(t_i), t_i) - f(h_{t_i}, t_i)|$$
$$+ \frac{M}{2}s^2. \quad (18)$$

Because the assumption of $f$ and its derivative being continuous and bounded, according to the mean value theorem, we have

$$f(h(t_i), t_i) - f(h_{t_i}, t_i) = f_h(h_{t_i}^*, t_i)e_{t_i}, \quad (19)$$

where $h_{t_i}^*$ is some number between $h(t_i)$ and $h_{t_i}$. Since $f_h$ is bounded, we have

$$|f(h(t_i), t_i) - f(h_{t_i}, t_i)| \leq R|e_{t_i}|, \quad (20)$$

where $R$ is some constant. By substituting Eq. 20 into Eq. 18, we obtain the relation between the errors of two consecutive steps:

$$|e_{t_{i+1}}| \leq (1 + sR)|e_{t_i}| + \frac{M}{2}s^2. \quad (21)$$

For simplicity, let $C = (1 + sR)$. We can then iterative apply the inequality starting from $t = t_0$ and $e_{t_0} = h(t_0) - ht_0 = 0$ as

$$|e_{t_1}| \leq \frac{M}{2}s^2,$$
$$|e_{t_2}| \leq (1 + C)\frac{M}{2}s^2,$$
$$\vdots$$
$$|e_{t_n}| \leq (1 + C + \cdots + C^{n-1})\frac{M}{2}s^2, \quad (22)$$

and since $(1 + C + \cdots + C^{n-1}) = \frac{1 - C^n}{1 - C} = \frac{(1 + Rs)^n - 1}{Rs}$, we then obtain the error bound of the approximated final result

$$|e_{t_n}| \leq \frac{(1 + Rs)^n - 1}{R}\frac{M}{2}s$$
$$\leq \frac{e^{Rns} - 1}{R}\frac{M}{2}s$$
$$\leq \frac{e^{RT} - 1}{R}\frac{M}{2}s \quad (ns = T). \quad (23)$$

Denoting $K = \frac{e^{RT} - 1}{R}\frac{M}{2}$, then the global truncation error is

$$|e_{t_n}| \leq Ks, \quad (24)$$

which is of order $O(s)$, i.e., linear to the step size.

| Hyper-Parameter | Value |
|---|---|
| lr | 1e-4 |
| lr decay style | linear |
| min lr | 1e-5 |
| iterations | 300,000 |
| batch size | 1024 |
| weight decay | 0.01 |
| warmup ratio | 0.01 |
| gradient norm | 1.0 |
| dropout | 0.0 |

Table 2: Hyper-parameters for pre-training all the BERT models in this work.

| Hyper-Parameter | Value |
|---|---|
| lr | 1e-4 |
| lr decay style | cosine |
| min lr | 1e-5 |
| iterations | 300,000 |
| batch size | 512 |
| weight decay | 0.01 |
| warmup ratio | 0.01 |
| gradient norm | 1.0 |
| dropout | 0.1 |

Table 3: Hyper-parameters for pre-training all the GPT-2 models in this work.

As for the difference between the final approximated state $\tilde{h}_T$ obtained using larger step size $\beta$ and the state $h_T$ obtained using the original step size $s$, since $|e_T| = |h_T - h(T)| \leq Ks$ and $|\tilde{e}_T| = |\tilde{h}_T - h(T)| \leq K\beta s$, therefore we have

$$|\tilde{h}_T - h_T| = \left| \left( \tilde{h}_T - h(T) \right) - (h_T - h(T)) \right|$$

$$= \left| \tilde{h}_T - h(T) \right| + |h_T - h(T)| \quad (25)$$

$$\leq K(1 + \beta)s. \quad (26)$$

Therefore the difference between $h_T$ and $\tilde{h}_T$ is also bounded, and is linear to step size $s$. When the step size at different time is different, one can easily derive that the global truncated error is bounded by the largest step size $\beta^*$

$$|\tilde{h}_T - h_T| \leq K(1 + \beta^*)s \quad (27)$$

## C   Additional Figures for Section 4.3 and Section 4.4

Due to the length constraint, we place the additional plots for Section 4.3 at Figs. 8 and 10, and plots for Section 4.4 at Fig. 9.

## D   Configurations for the Pre-training

We use Megatron-LM (Narayanan et al., 2021) as the framework to pre-train our parameter-shared BERT and GPT-2. The OpenWebText dataset for pre-training GPT-2 is prepared following the instructions in Megatron-LM repository[1], and the preparation of the Pile dataset for pre-training BERT follows the instructions in the Megatron-Deepspeed repository[2]. We use

---

[1] https://github.com/NVIDIA/Megatron-LM
[2] https://github.com/microsoft/Megatron-DeepSpeed

AdamW (Loshchilov and Hutter, 2019) as the optimizer. The model configurations for our BERT and GPT-2 are kept the same as BERT$_{\text{large}}$ and GPT-2$_{\text{large}}$ respectively.

## E   Configurations for the Downstream Tasks

We adopt the approach employed by Megatron-LM framework for handling MNLI and RACE tasks. For the classification tasks MNLI and SST-2, we utilize the hidden state of the [CLS] token for classification and report accuracy on the development set. In the RACE task, we predict the probability of each answer using the [CLS] token's representation and report test set accuracies. Regarding the SQuAD v1.1 and v2.0 tasks, we adhere to BERT's training procedure, applying a span extraction loss, and record the F1 score on the development set using the official evaluation script[3].

During fine-tuning BERT on all the downstream tasks, we use the linear learning rate warmup and decay schedule. The gradient norm is constrained to 1.0. We apply a dropout rate of 0.1 and a weight decay of 0.01. Further hyperparameter details are documented in Table 4.

For GPT tasks, we adopt a zero-shot approach. The performance on the LAMBADA task is assessed using cloze accuracy, which involves predicting the last word (not the last token) based on the preceding tokens. Performance on Wikitext-103 is measured using the perplexity metric on the test set.

---

[3] https://github.com/rajpurkar/SQuAD-explorer

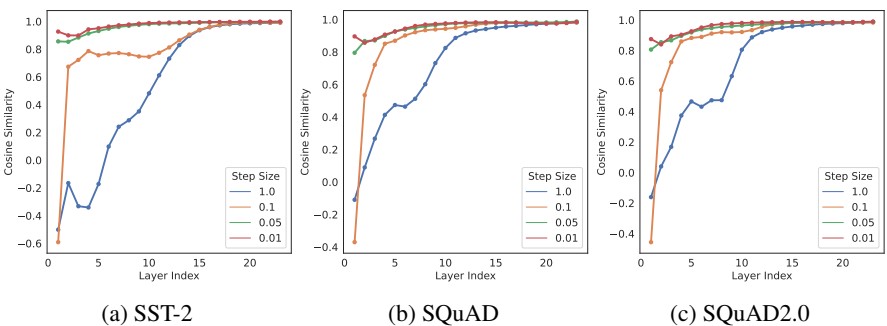

(a) SST-2        (b) SQuAD        (c) SQuAD2.0

Figure 8: The cosine similarity between the derivatives given by the model at two consecutive iterations.

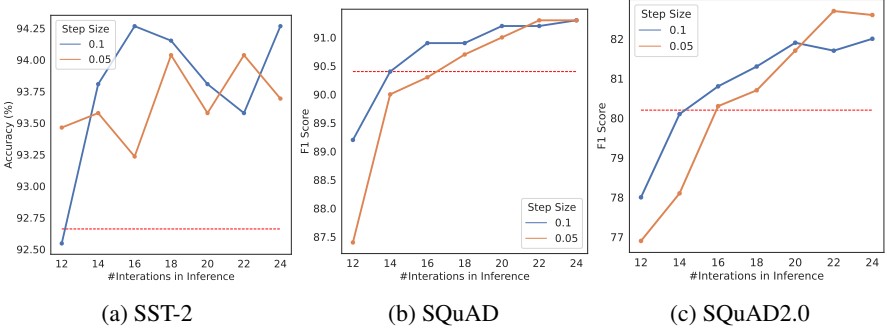

(a) SST-2        (b) SQuAD        (c) SQuAD2.0

Figure 9: The inference performance of partially-shared models with $n = 12$ sets parameters on SST-2, SQuAD and SQuAD2.0. The red dashed line represents the performance of the unshared 12 layer model.

## F  Configurations for the Early Exit

We employ the methodology from DeeBERT (Xin et al., 2020) for the early exit experiment. During the training phase, we keep the BERT model and the original classification head fixed and only train the additional classifiers on each layer. The training loss is computed as the sum of the losses from all additional classifiers.

In the inference phase, we calculate the entropy of the output logits at each layer. If the entropy value falls below a predetermined threshold at a layer, we halt the computation and take the prediction at this layer as the final prediction. We record the average number of iterations at the point of output across all instances. The thresholds utilised in this experiment are as follows: [0, 0.01, 0.05, 0.07, 0.1, 0.2, 0.3, 0.4, 0.5]. Higher thresholds induce an earlier exit.

## G  Performance under $n = 24$ Setting

In this section, we extend our partially-shared model to include $n = 24$ sets of parameters, which renders it equivalent to an unshared 24-layer model. The only difference is that we use the same initialization for the 24 sets of parameters. This attempt is only made on the GPT-2$_{\text{large}}$ model, and the cor-

responding results are presented in Table 5.

On close examination, we note a peculiar trend: as the iteration count reduces from 24 to 12, the zero-shot perplexity on the Wikitext-103 dataset first increases, and then decreases. This anomaly could be attributed to the model's inability to learn the usage of linear interpolation between the endpoint parameters for derivative calculation when $n = 24$. That is, when $n = 24$, the linear interpolation (Eq. 6) always returns the parameters on one of the endpoint. Consequently, during the inference phase, as we increase the step size, the parameters derived through linear interpolation start deviating from the parameters utilized by the model during training. This divergence is potentially responsible for a significant degradation in the model's performance.

Interestingly, when the scaling factor $\beta$ for each iteration is adjusted to 2 and the iteration count is reduced to 12, the model yields a reasonable performance. We hypothesize this is due to the fact that when scaling factors are integers, the parameters derived remain endpoint parameters, which the model has been accustomed to handle during the training phase. However, the experiment still highlight the potential of our method when applying to the unshared model.

| Hyper-Parameter | MNLI | SST-2 | RACE | SQuAD | SQuAD2 |
|---|---|---|---|---|---|
| Step Size = 1 | | | | | |
| lr | 2e-05 | 1e-05 | 1e-05 | 3e-05 | 2e-05 |
| epochs | 5 | 5 | 5 | 3 | 5 |
| batch size | 64 | 32 | 16 | 32 | 32 |
| warmup ratio | 0.05 | 0.01 | 0.1 | 0.01 | 0.01 |
| Step Size = 0.1 | | | | | |
| lr | 1e-05 | 1e-05 | 2e-05 | 3e-05 | 2e-05 |
| epochs | 5 | 5 | 5 | 3 | 5 |
| batch size | 16 | 32 | 16 | 32 | 32 |
| warmup ratio | 0.1 | 0.1 | 0.05 | 0.1 | 0.1 |
| Step Size = 0.05 | | | | | |
| lr | 1e-05 | 3e-05 | 2e-05 | 3e-05 | 2e-05 |
| epochs | 5 | 5 | 5 | 3 | 5 |
| batch size | 16 | 32 | 32 | 32 | 16 |
| warmup ratio | 0.1 | 0.05 | 0.1 | 0.01 | 0.01 |
| Step Size = 0.01 | | | | | |
| lr | 4e-05 | 2e-05 | 3e-05 | 2e-05 | 3e-05 |
| epochs | 5 | 5 | 5 | 3 | 5 |
| batch size | 64 | 16 | 16 | 16 | 32 |
| warmup ratio | 0.05 | 0.01 | 0.01 | 0.05 | 0.01 |

Table 4: Hyper-parameters for downstream tasks with different step sizes.

| #Iters | PPL |
|---|---|
| 24 | 19.74 |
| 20 | 34.67 |
| 16 | 168.68 |
| 14 | 77.21 |
| 12 | 57.93 |

Table 5: Zero-shot perplexity on Wikitext-103 of the partially-shared model with $n = 24$.

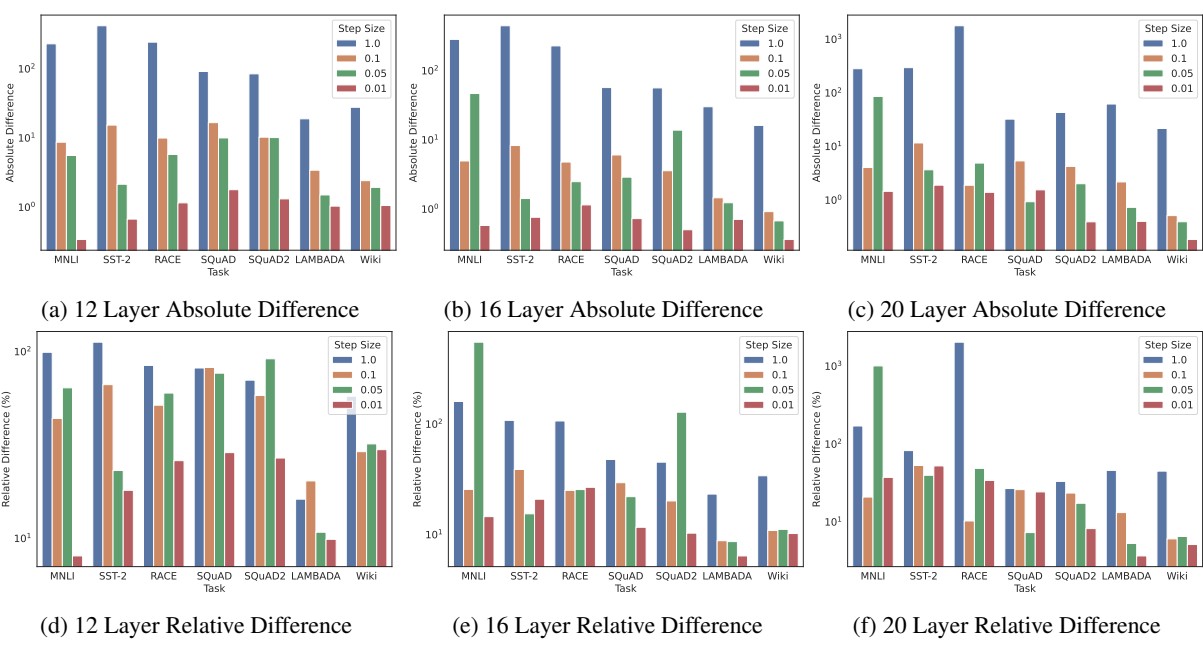

(a) 12 Layer Absolute Difference     (b) 16 Layer Absolute Difference     (c) 20 Layer Absolute Difference

(d) 12 Layer Relative Difference     (e) 16 Layer Relative Difference     (f) 20 Layer Relative Difference

Figure 10: The absolute and relative difference between the final hidden states obtained with 24 iterations and 12, 16, 20 iterations.