# OpenReview forum: "Boosting Inference Efficiency: Unleashing the Power of Parameter-Shared Pre-trained Language Models"
_EMNLP/2023/Conference — EMNLP 2023 Findings_

### Official Review · Reviewer_y9YL · 2023-07-30

**Soundness:** 2

**Excitement:**

3: Ambivalent: It has merits (e.g., it reports state-of-the-art results, the idea is nice), but there are key weaknesses (e.g., it describes incremental work), and it can significantly benefit from another round of revision. However, I won't object to accepting it if my co-reviewers champion it.

**Paper Topic And Main Contributions:**

In these days, the large-scale pre-trained language models such as GPT series have succeeded in various NLP applications. In this paradigm, to save the computational costs, we have to explore an efficient method. As a parameter-efficient method, the parameter sharing approach is widely used. In this paper, the authors propose a method to accelerate inference speed of the parameter sharing Transformers. The authors indicated that we can accelerate the inference speed of Transformers whose parameters are shared with all layers by modifying weights to each sub-layers such as FFNs and attention modules. The authors focused on BERT and GPT-2, and conducted experiments on several tasks.

**Questions For The Authors:**

s=0.05 was consistently better than s=0.1 in the fully-shared setting but s=0.1 outperformed s=0.05 in most datasets in the partially shared setting. Do the authors have any intuition on this discrepancy?

**Reasons To Accept:**

This paper addresses an important problem, i.e., the efficiency of Transformers. The proposed method, which applies weights to sub-layers during the inference, is an interesting approach to accelerates the inference speed.

**Reasons To Reject:**

This paper is inadequate to publish due to some reasons.

In my understanding, the claims of this paper have not been proved yet. The authors varied beta during the experiments but did not report the results in fixed beta such as beta=1. Thus, it is difficult to agree with the claim "we can accelerate the inference speed by using appropriate beta." In addition, I recommend reporting the portability of beta. Can we use beta which is adjusted to a specific task for other tasks?

In addition, the authors should compare the proposed method with the early exit techniques. These method might need extra modules and/or computational costs (especially, during the training) but the proposed method also need additional computational costs to search the appropriate beta before the inference. The proposed method is more efficient in the total cost?

This paper is hard to follow. At least, the authors should reduce the ambiguity in notations. For example, in Equation (3), f_t -> f_i in my understanding. In addition, the authors defined the number of layers as T in Section 2 but use L in Section 3.3. Moreover, the description contains several errors. For example, in Figure 4, vertical axes of (a), (b), and (c) represent perplexity but the correct representation is Accuracy. Moreover, the authors mentioned figures in Appendices for the main discussion such as Section 4.3.2. The authors should include them into the main part if the authors use them to support the main claims.

**Reproducibility:**

2: Would be hard pressed to reproduce the results. The contribution depends on data that are simply not available outside the author's institution or consortium; not enough details are provided.

**Reviewer Confidence:**

3: Pretty sure, but there's a chance I missed something. Although I have a good feel for this area in general, I did not carefully check the paper's details, e.g., the math, experimental design, or novelty.

---

> ### Author Rebuttal · Authors · 2023-08-27
>
> Thank you for your thorough and constructive review of our paper. We sincerely appreciate the time and effort you've put into pointing out the weaknesses. Below, we will respond to each of your comments.
>
> ---
>
> **Question 1**: The results of fixed $\beta$ (the step-size scaling factor) are not reported.
>
> **Answer 1**: We initially aimed to demonstrate the maximum potential of acceleration via our method by exploring the best $\beta$. **In fact, the search is very fast, for example, searching for the $\beta$ value we reported on SST-2 on a single GPU takes only about 14 minutes.** Nonetheless, you rightly pointed out the importance of reporting results with a fixed $\beta$. As such, we now present results using a direct setting of $\beta=24/\text{reduced-iteration}$ for various iteration counts:
>
> `fully-shared step size=1.0`
>
> | #Iters | MNLI               | SST-2             | RACE               | SQuAD              | SQuAD2             | Wiki-103             | LAMBADA            |
> | ------ | ------------------ | ----------------- | ------------------ | ------------------ | ------------------ | -------------------- | ------------------ |
> | 24     | 83.6             | 91.1            | 64.3             | 90.2             | 81.5             | 33.0               | 31.1             |
> | 20     | 83.2 (-0.5\%)  | 90.6 (-0.5\%) | 63.5 (-1.2\%)  | 88.8 (-1.6\%)  | 75.7 (-7.1\%)  | 38.3 (+16.1\%)   | 16.1 (-48.2\%) |
> | 16     | 81.0 (-3.1\%)  | 89.5 (-1.8\%) | 60.1 (-6.5\%)  | 82.6 (-8.4\%)  | 57.2 (-29.8\%) | 64.7 (+96.1\%)   | 6.4 (-79.4\%)  |
> | 12     | 71.9 (-14.0\%) | 86.6 (-4.9\%) | 52.3 (-18.7\%) | 61.7 (-31.6\%) | 33.7 (-58.7\%) | 168.5 (+410.6\%) | 1.6 (-94.9\%)  |
>
> `fully-shared step size=0.1`
>
> | #Iters | MNLI               | SST-2              | RACE               | SQuAD              | SQuAD2             | Wiki-103           | LAMBADA            |
> | ------ | ------------------ | ------------------ | ------------------ | ------------------ | ------------------ | ------------------ | ------------------ |
> | 24     | 83.2             | 91.4             | 67.5             | 90.1             | 81.4             | 33.4             | 25.5             |
> | 20     | 83.6 (+0.5\%)  | 91.6 (+0.2\%)  | 67.3 (-0.3\%)  | 89.5 (-0.7\%)  | 79.8 (-2.0\%)  | 34.9 (+4.5\%)  | 23.5 (-7.8\%)  |
> | 16     | 82.6 (-0.7\%)  | 89.6 (-2.0\%)  | 61.3 (-9.2\%)  | 83.5 (-7.3\%)  | 66.9 (-17.8\%) | 35.4 (+6.0\%)  | 23.7 (-7.1\%)  |
> | 12     | 52.2 (-37.3\%) | 79.4 (-13.1\%) | 44.1 (-34.7\%) | 33.5 (-62.8\%) | 36.9 (-54.7\%) | 45.4 (+35.9\%) | 18.1 (-29.0\%) |
>
> `fully-shared step size=0.05`
>
> | #Iters | MNLI              | SST-2             | RACE              | SQuAD             | SQuAD2            | Wiki-103           | LAMBADA            |
> | ------ | ----------------- | ----------------- | ----------------- | ----------------- | ----------------- | ------------------ | ------------------ |
> | 24     | 82.6            | 90.9            | 68.1            | 90.3            | 81.3            | 33.9             | 26.5             |
> | 20     | 82.8 (+0.2\%) | 91.2 (+0.3\%) | 68.3 (+0.3\%) | 90.2 (-0.1\%) | 81.3 (+0.0\%) | 35.2 (+3.8\%)  | 27.4 (+3.4\%)  |
> | 16     | 83.2 (+0.7\%) | 91.1 (+0.2\%) | 68.2 (+0.1\%) | 90.0 (-0.3\%) | 80.8 (-0.6\%) | 35.5 (+4.7\%)  | 27.5 (+3.8\%)  |
> | 12     | 81.7 (-1.1\%) | 89.5 (-1.5\%) | 62.2 (-8.7\%) | 85.5 (-5.3\%) | 74.5 (-8.4\%) | 45.6 (+34.5\%) | 21.9 (-17.4\%) |
>
> `fully-shared step size=0.01`
>
> | #Iters | MNLI              | SST-2             | RACE              | SQuAD             | SQuAD2            | Wiki-103           | LAMBADA           |
> | ------ | ----------------- | ----------------- | ----------------- | ----------------- | ----------------- | ------------------ | ----------------- |
> | 24     | 81.7            | 90.1            | 64.4            | 89.3            | 79.5            | 35.0             | 27.1            |
> | 20     | 81.8 (+0.1\%) | 90.5 (+0.4\%) | 64.7 (+0.5\%) | 89.3 (+0.0\%) | 79.6 (+0.1\%) | 35.9 (+2.6\%)  | 27.0 (-0.4\%) |
> | 16     | 82.0 (+0.4\%) | 90.3 (+0.2\%) | 64.0 (-0.6\%) | 89.2 (-0.1\%) | 79.5 (+0.0\%) | 36.0 (+2.9\%)  | 27.8 (+2.6\%) |
> | 12     | 82.0 (+0.4\%) | 89.7 (-0.4\%) | 60.8 (-5.6\%) | 83.2 (-6.8\%) | 73.1 (-8.1\%) | 45.1 (+28.9\%) | 25.6 (-5.5\%) |
>
> For the fully-shared model, the results indicate that directly setting $beta$ to $24/\text{reduced-iteration}$ can produce favorable outcomes without searching. However, for models with a step size of 0.1, this approach seems to be not so effective. **Nonetheless, the significance of the zero-shot acceleration achieved in models with step sizes of 1, 0.05, and 0.01 should not be overlooked. We anticipate that these insights might pave the way for more in-depth research in model inference acceleration.** Also, with a simple and quick search, we can yield much better $\beta$ values as reported in our paper. These observations are consistent for the fully-shared model as well:
>
> `fully-shared step size=0.1`
>
> | #Iters | MNLI              | SST-2             | RACE               | SQuAD              | SQuAD2             | Wiki-103            | LAMBADA            |
> | ------ | ----------------- | ----------------- | ------------------ | ------------------ | ------------------ | ------------------- | ------------------ |
> | 24     | 85.7            | 93.1            | 71.7             | 91.2             | 82.0             | 21.3              | 39.0             |
> | 20     | 84.9 (-0.9\%) | 93.2 (+0.1\%) | 70.5 (-1.7\%)  | 90.6 (-0.7\%)  | 80.2 (-2.2\%)  | 31.1 (+46.0\%)  | 31.3 (-19.7\%) |
> | 16     | 84.7 (-1.2\%) | 92.2 (-1.0\%) | 69.1 (-3.6\%)  | 89.7 (-1.6\%)  | 78.2 (-4.6\%)  | 24.8 (+16.4\%)  | 27.9 (-28.5\%) |
> | 12     | 80.7 (-5.8\%) | 90.6 (-2.7\%) | 59.8 (-16.6\%) | 78.3 (-14.1\%) | 67.5 (-17.7\%) | 47.4 (+122.5\%) | 28.0 (-28.2\%) |
>
> `fully-shared step size=0.05`
>
> | #Iters | MNLI              | SST-2             | RACE               | SQuAD              | SQuAD2             | Wiki-103            | LAMBADA            |
> | ------ | ----------------- | ----------------- | ------------------ | ------------------ | ------------------ | ------------------- | ------------------ |
> | 24     | 85.2            | 93.3            | 71.4             | 91.4             | 82.6             | 22.1              | 39.2             |
> | 20     | 84.7 (-0.6\%) | 92.9 (-0.4\%) | 68.4 (-4.2\%)  | 90.1 (-1.4\%)  | 80.7 (-2.3\%)  | 35.6 (+61.1\%)  | 35.8 (-8.7\%)  |
> | 16     | 84.4 (-0.9\%) | 92.7 (-0.6\%) | 68.2 (-4.5\%)  | 89.5 (-2.1\%)  | 76.2 (-7.7\%)  | 26.5 (+19.9\%)  | 38.0 (-3.1\%)  |
> | 12     | 78.7 (-7.6\%) | 87.5 (-6.2\%) | 54.9 (-23.1\%) | 73.6 (-19.5\%) | 63.8 (-22.8\%) | 52.9 (+139.4\%) | 25.3 (-35.5\%) |
>
> For the fully-shared model, directly setting the beta value yields slightly inferior results, but at 16 layers, the performance remains quite good. **We believe our findings, particularly the zero-shot acceleration displayed in our experiments, will stimulate further interest and research in the realm of model inference acceleration.**
>
> ---
>
> **Question 2**: Lack of comparison with early exit techniques.
>
> **Answer 2**: We discuss the relationship between our method and early exit techniques in Section 4.5. **Our approach is orthogonal to early exit techniques, complementing each other.** Our experiments have proven that when they are combined, even superior inference acceleration can be attained (Figure 7). Considering our method's zero-shot nature, it can be viewed as an enhancement to pre-existing early exit techniques. Moreover, **since early exit techniques require training on the training dataset and introduce additional new parameters, directly comparing our method with them might not be fair.**
>
> ---
>
> **Question 3**: Notational inconsistencies and paper readability issues.
>
> **Answer 3**: Thank you for your meticulous feedback regarding the inconsistencies and errors. We are committed to resolving all the highlighted issues to offer a clearer and more reader-friendly paper. Regarding the issue where we referenced figures from the appendix in the main text, we will adjust the layout to move essential content to the main body. Thank you once again for your patient correction!
>
> ---
>
> **Question 4**: In the fully-shared model, $s=0.05$ performs better, but in the partially-shared model, $s=0.1$ is better. Is there any explanation?
>
> **Answer 4**: The observed discrepancy indeed presents an intriguing aspect of our experiments. A potential hypothesis could be that the fully-shared model, with each iteration influencing the same parameters, has a larger gradient norm than the partially-shared model. As such, using a smaller step size might be analogous to gradient norm clipping, thereby reducing the gradient variation. However, this remains a hypothesis, and we are actively exploring more insights into this phenomenon.
>
> ---
>
> The proposed revisions, guided by your invaluable feedback, will significantly enhance the paper. We're grateful for your insights and look forward to furthering this discussion.

---

### Official Review · Reviewer_T3zw · 2023-08-05

**Soundness:** 3

**Excitement:**

3: Ambivalent: It has merits (e.g., it reports state-of-the-art results, the idea is nice), but there are key weaknesses (e.g., it describes incremental work), and it can significantly benefit from another round of revision. However, I won't object to accepting it if my co-reviewers champion it.

**Paper Topic And Main Contributions:**

- The paper presents a neural ordinary differential equation (ODE) inspired technique aimed at enhancing the inference efficiency of parameter-shared PLMs (PSPLM).
- Furthermore, a novel pre-training method, involving a slightly altered forward propagation, is proposed to further accelerate the inference of PSPLM.
- Additionally, the study showcases the potential of the proposed strategy in the context of partially-shared PLM, highlighting the flexibility and robustness of the approach.
- Overall, the findings contribute to the advancement of PSPLM research and underscore the efficiency gains achievable through the proposed techniques.

**Questions For The Authors:**

A: What is the best practive for determining the step size and number of iterations? While it may vary depending on the model or data, even suggesting heuristic methods to find appropriate values would be valuable.

**Reasons To Accept:**

- There is definite merit in reducing the inference cost without the need for additional modules.
- The paper holds value in the presence of several insightful empirical results.

**Reasons To Reject:**

- It is disappointing that there is no method provided to find appropriate values for important variables such as step size or number of iterations, leaving heavy reliance on heuristics.
- Additionally, it would be beneficial to include an analysis of the model's scale. Given the recent attention towards LLMs that demonstrate diverse generalization abilities, showing the potential effectiveness of this methodology across various scales and applicability in LLMs would enhance the significance of the research.
- As the authors also acknowledge in the limitations section, further validation is required for Partially-shared PLMs.

**Reproducibility:**

3: Could reproduce the results with some difficulty. The settings of parameters are underspecified or subjectively determined; the training/evaluation data are not widely available.

**Reviewer Confidence:**

3: Pretty sure, but there's a chance I missed something. Although I have a good feel for this area in general, I did not carefully check the paper's details, e.g., the math, experimental design, or novelty.

---

> ### Author Rebuttal · Authors · 2023-08-27
>
> We sincerely thank you for the thorough review of our paper! We value the feedback provided and have made efforts to address your concerns. Here are our responses:
>
> ---
>
> **Question 1**: Best practice for determining the step size and number of iterations?
>
> **Answer 1**: **The number of iterations is determined based on actual needs when using the model for inference**. The user of the model decides on the number of iterations based on the complexity of the problem, the resources they have, and their tolerance for errors. Fewer iterations lead to faster inference but potentially worse results. **As for the step size during pre-training, this is a hyperparameter, and it's generally challenging to systematically determine the optimal value of a hyperparameter.** Our empirical results suggest step sizes between 0.5 to 0.05 work well. For the step size scaling factor $\beta$ during inference, the reported values in our paper are derived using Optuna toolkit searches on each task's dev set (line 303). We searched for 50 trials. **It is really fast since only one round of inference on the dev set is required in each trial, and it can be easily parallelized.** For example, when using one GPU, the search only takes around 14 minutes for SST-2. Additionally, as shown in our response to reviewer y9YL, we give evidence that setting the beta value directly to $\beta = 24 / \text{reduced-iteration}$ also yields satisfactory results.
>
> ---
>
> **Question 2**: Effectiveness of this method across scales.
>
> **Answer 2**: Due to rebuttal time constraints, we don't have enough resources to train a larger model. As an alternative, we train two parameter-shared BERT base and reported their performance (without searching $\beta$) on MNLI as follows:
>
> | #Iters | MNLI (step size=0.1) | MNLI (step size=0.05) |
> | ------ | -------------------- | --------------------- |
> | 12     | 81.7               | 81.7                |
> | 10     | 81.9 (+0.2\%)    | 82.0 (+0.4\%)     |
> | 8      | 81.9 (+0.2\%)    | 82.1 (+0.5\%)     |
> | 6      | 79.8 (-2.3\%)    | 79.8 (-2.3\%)     |
>
> After reducing the number of iterations, we directly set $\beta$ to $12/\text{reduced-iter}$. As can be seen, increasing the step size after reducing the number of iterations is also effective for a small model at the BERT base level . As for larger models, we believe our approach will still be applicable. Although we haven't tested our method on billion-parameter-level models, we believe our approach has been validated in practical models like BERT and has practical value, making it worthy of publication. This also paves a potential new way for accelerating model inference in the future.
>
> ---
>
> We hope our responses address your concern. Thank you again for your patient review!

---

### Official Review · Reviewer_k26S · 2023-08-07

**Paper Topic And Main Contributions:** 1. The paper presents a novel way of …
**Typos Grammar Style And Presentation Improvements:** 1. It would be good to bring at least…
**Soundness:** 4

**Excitement:**

4: Strong: This paper deepens the understanding of some phenomenon or lowers the barriers to an existing research direction.

**Questions For The Authors:**

Question A: How are the beta values tuned for each task ? Is it on the dev set ?
Question B: For the results in Table 1, it seems like encoder only models are more robust to reduced iterations compared to zero shot performance from decoder only models. Is there any intuition on why that might be the case ?
Question C: Is there an intuition on why models trained with a smaller step-size outperform their size=1 counterparts ?
Question D: From Figure 10 (b), the absolute drop between h_{T} and \tilde{h}_{T} between Race and SQuAD is about the same and quite less. Based on the hypothesis presented in the paper, this should imply that the downstream performance between the two should be quite similar too. However, the downstream performance is quite different for the case of RACE, with the 16 iteration model outperforming the 24 iteration model. Is there an intuition on why that might be the case ?


**Reasons To Accept:**

1. Leveraging the connections of Pre-LayerNorm transformer networks to ODEs for reducing inference latency by dynamically scaling step sizes is, to the best of my knowledge, novel and of practical utility.
2. The inference time step-size scaling results are quite compelling, especially for encoder only models
3. The low step-size training experiments are theoretically motivated, and the paper presents strong experiments validating the claims (Section 4.3.2)
4. The proposed method for partially shared PLMs is very interesting with compelling results, and demonstrates the possibility of additional future explorations.
5. The integration with early exit strategies further demonstrates the utility of the presented techniques.

**Reasons To Reject:**

1. Some results need additional context to better understand the implications, in my opinion. Specifically, for the section on partial sharing, in addition to Figure 6, it would also be good to
1.1 Present the upper-bound performance using a 24 layer unshared model
1.2 Present the iteration speed, as done in Table 1.
Since the paper is mostly presented as a method for improving speed, an iso-parameter unshared model with lower performance somewhat deviates from the main message (since in this case, as per my understanding, the iso-parameter model would technically have lower latency). While line 495 partially addresses this (w.r.t comparing the performance of a post reduction model comparing to baseline), I think the above two data-points are valuable to get a full picture.

2. The details about how the beta values are chosen per iteration budget are somewhat fuzzy. Specifically, what is the set that the beta values are tuned on ? For the language model setup, where metrics are zero-shot, are the beta values tuned at a per task level, or at an aggregate level ? This is important since for decoder only language models that operate on a zero-shot setup, one of the primary advantages is the task agnostic nature of the models. Thus it would be good to see how robust the approach is to task agnostic tuning (eg: tuning on perplexity on the validation set of the pre-training objective), and how that might translate to zero-shot setups. On a similar note, it would be good to understand how robust the approach is to different sets of beta values.

3. Appendix G presents an analysis based on Table 5. However at a perplexity of >= 35, the model performance is really bad. So any conclusions / trends based on that might not be very robust.



**Reproducibility:**

4: Could mostly reproduce the results, but there may be some variation because of sample variance or minor variations in their interpretation of the protocol or method.

**Reviewer Confidence:**

3: Pretty sure, but there's a chance I missed something. Although I have a good feel for this area in general, I did not carefully check the paper's details, e.g., the math, experimental design, or novelty.

---

> ### Author Rebuttal · Authors · 2023-08-27
>
> We greatly appreciate your comprehensive review and insightful feedback on our work. We are gratified by your positive assessment and will address each of your concerns in sequence.
>
> ---
>
> **Question 1**: Tuning of beta values (scaling factor of the step size) and use of the dev set.
>
> **Answer 1**: We concur with your viewpoint on the importance of specifying the process. **For tasks with a dev set, such as RACE, we tune beta values using dev set and report the performance on test set. For GLUE datasets, we follow conventions of prior works, tuning on the dev set and reporting the dev result.** We utilized the Optuna toolkit (line 303), running 50 trials for each task. It is really fast since only one round of inference on the dev set is required in each trial, and it can also be easily parallelized. For example, when using one GPU, the search only takes around 14 minutes for SST-2. In our response to reviewer y9YL, we also show that we can still achieve good results without tuning the beta value and setting it directly to $\beta = 24 / \text{reduced-iteration}$. **We have shown the potential of varying the step size at each step in our paper, and we believe that continuing to explore how to quickly determine the beta values would be an important and practical direction.**
>
> ---
>
> **Question 2**: Intuition on why encoder-only models are more robust than decoder-only models?
>
> **Answer 2**: **It's possible the perceived fragility of decoder-only models isn't intrinsic but rather a result of the evaluation metrics.** On wikitext-103, we used the perplexity metric which can degrade exponentially with diminishing model capability. The larger classification space in text generation (the entire vocabulary) versus encoder tasks might also explain the pronounced decline in LAMBADA accuracy.
>
> ---
>
> **Question 3**: Intuition on why models trained with a smaller step size outperform their size=1 counterparts?
>
> **Answer 3**: We hypothesize that smaller step sizes produce smaller gradients, which might mitigate issues like large gradient variance during training. This could act in a manner reminiscent of gradient norm clipping. Your observation is astute and warrants further investigation.
>
> ---
>
> **Question 4**: Is there an intuitive explanation for the difference in downstream performance between the 16 iteration and 24 iteration models for RACE, despite the similar absolute drop observed between $h_{T}$ and $\tilde{h}_{T}$ in Figure 10 (b)?
>
> **Answer4**: We can truly feel your meticulousness in reviewing our paper from this question, thank you! **Comparing absolute drops in $h_{T}$ and $\tilde{h}_{T}$ across distinct tasks might not provide meaningful insights due to different loss landscapes** (or accuracy landscapes if it's a classification task). For example, even minor deviations from optimum values can cause sharp performance declines in steeper landscapes, whereas in flatter landscapes, the same deviations might result in milder effects. The absolute difference only makes sense for the same task.
>
> ---
>
> Regarding the suggestions mentioned in the reasons to reject, such as the performance and speed of the unshared 24-layer model, we will incorporate them into our paper for a more comprehensive presentation. And we will correct all the typos that you have patiently pointed out. Again, thank you for your comprehensive review and feedback! We hope our clarifications have addressed your concerns.

---

### Meta-Review · Area_Chair_tT6L · 2023-09-08

**Recommendation:** 3

**Metareview:**

This paper presents methods for improving the inference speed for parameter shared pre-trained language models, presenting both theoretical and empirical evidence. The reviewers appreciate the importance of the problem, the novelty, elegance, practicality and theoretical foundations, the richness of the results, and their quality. In contrast, there were initially claims about missing experiments and comparisons, which resulted in unjustified claims, and clarity issues. Post-rebuttal, many of these claims seem to be resolved (as evident in the internal discussion amongst the reviewers), but the clarity issues are still very much concerning to some of them. The scaling concerns also still remain, though I have discounted them due to the ACL policy (https://www.aclweb.org/portal/content/efficient-nlp-policy-document) which discourages unjustified requests for larger experiments.

---

### Decision · Program_Chairs · 2023-10-07

**Decision:**

Accept-Findings

**Comment:**

This paper presents methods for improving the inference speed for parameter shared pre-trained language models, presenting both theoretical and empirical evidence. The reviewers appreciate the importance of the problem, the novelty, elegance, practicality and theoretical foundations, the richness of the results, and their quality. In contrast, there were initially claims about missing experiments and comparisons, which resulted in unjustified claims, and clarity issues. Post-rebuttal, many of these claims seem to be resolved (as evident in the internal discussion amongst the reviewers), but the clarity issues are still very much concerning to some of them. The scaling concerns also still remain, though I have discounted them due to the ACL policy (https://www.aclweb.org/portal/content/efficient-nlp-policy-document) which discourages unjustified requests for larger experiments.